# Efficacy and Safety of Transcranial Electric Stimulation during the Perinatal Period: A Systematic Literature Review and Three Case Reports

**DOI:** 10.3390/jcm11144048

**Published:** 2022-07-13

**Authors:** Andrew Laurin, Noémie Nard, Marine Dalmont, Samuel Bulteau, Cloé Bénard, Olivier Bonnot, Norbert Winer, Françoise Dupont, Gisèle Apter, Frédérique Terranova-Commessie, Olivier Guillin, Wissam El-Hage, Anne Sauvaget, Maud Rothärmel

**Affiliations:** 1Movement-Interactions-Performance (MIP), UR 4334, Nantes Université, CHU Nantes, F-44000 Nantes, France; anne.sauvaget@chu-nantes.fr; 2Service de Psychiatrie G08 & Service Hospitalo-Universitaire de Psychiatrie de l’Adulte, Centre Hospitalier du Rouvray, 4, rue Paul Eluard, F-76300 Sotteville-lès-Rouen, France; noemie.nard@ch-lerouvray.fr (N.N.); marine.dalmont@ch-lerouvray.fr (M.D.); frederique.terranovacommessie@ch-lerouvray.fr (F.T.-C.); olivier.guillin@ch-lerouvray.fr (O.G.); maud.rotharmel@ch-lerouvray.fr (M.R.); 3Faculté de Médecine, Université de Normandie, 22, boulevard Gambetta, F-76000 Rouen, France; 4Clinical Investigation Unit 18, Department of Addictology and Psychiatry, CHU Nantes, F-44000 Nantes, France; samuel.bulteau@chu-nantes.fr; 5MethodS in Patients-Centered Outcomes and HEalth Research (SPHERE), Institut National de la Santé et de la Recherche Médicale (INSERM), Nantes Université, F-44000 Nantes, France; 6Nantes Université, F-44000 Nantes, France; cloe.benard@etu.univ-nantes.fr (C.B.); olivier.bonnot@chu-nantes.fr (O.B.); 7Service de Gynécologie Obstétrique, Institut National de Recherche pour l’Agriculture, l’Alimentation et l’Environnement (INRAE[M1]), UMR 1280, PhAN, Nantes Université, CHU Nantes, F-44000 Nantes, France; norbert.winer@chu-nantes.fr; 8CHU Nantes, F-44000 Nantes, France; francoise.dupont@chu-nantes.fr; 9Service de Pédopsychiatrie Universitaire, Groupe Hospitalier du Havre, F-76290 Havre, France; gisele.apter@ch-havre.fr; 10Faculté de Médecine de Normandie, 22, boulevard Gambetta, F-76000 Rouen, France; 11UMR 1253, iBrain, INSERM, Université de Tours, F-37000 Tours, France; wissam.elhage@univ-tours.fr

**Keywords:** transcranial direct current stimulation, transcranial alternating current stimulation, pregnancy, perinatal period, postpartum period, breastfeeding

## Abstract

**Introduction**: The perinatal period is an at-risk period for the emergence or decompensation of psychiatric disorders. Transcranial electrical stimulation (tES) is an effective and safe treatment for many psychiatric disorders. Given the reluctance to use pharmacological treatments during pregnancy or breastfeeding, tES may be an interesting treatment to consider. Our study aims to evaluate the efficacy and safety of tES in the perinatal period through a systematic literature review followed by three original case reports. **Method**: Following PRISMA guidelines, a systematic review of MEDLINE and ScienceDirect was undertaken to identify studies on tES on women during the perinatal period. The initial research was conducted until 31 December 2021 and search terms included: tDCS, transcranial direct current stimulation, tACS, transcranial alternating current stimulation, tRNS, transcranial random noise stimulation, pregnancy, perinatal, postnatal, and postpartum. **Results**: Seven studies reporting on 33 women during the perinatal period met the eligibility criteria. No serious adverse effects for the mother or child were reported. Data were limited to the use of tES during pregnancy in patients with schizophrenia or unipolar depression. In addition, we reported three original case reports illustrating the efficacy and safety of tDCS: in a pregnant woman with bipolar depression, in a pregnant woman with post-traumatic stress disorder (sham tDCS), and in a breastfeeding woman with postpartum depression. **Conclusions**: The results are encouraging, making tES a potentially safe and effective treatment in the perinatal period. Larger studies are needed to confirm these initial results, and any adverse effects on the mother or child should be reported. In addition, research perspectives on the medico-economic benefits of tES, and its realization at home, are to be investigated in the future.

## 1. Introduction

The perinatal period, from pregnancy to one year after delivery, is a period of physical, biological, emotional, and psycho-social transformation in a woman’s life, imposing a significant adaptation that exposes her to a higher risk of acute psychiatric disorders [1]. Mood disorders are the most frequent psychiatric disorders in the perinatal period [2,3]. Indeed, depression has an incidence of about 10% during pregnancy and ranges between 5% and 25% in the postnatal period [4]. In women with bipolar disorder, the incidence of postpartum depressive disorder is estimated at 37%, with an increased risk of postpartum psychosis [5,6]. The presence of an untreated or poorly treated psychiatric disorder in the perinatal period is associated with an increased risk of poor pregnancy follow-up [7], associated with major somatic risks such as high blood pressure with a potential increased risk of pre-eclampsia [8,9], but also substance use disorder [10], suicidality [11], complications in fetal development [12], paternal depression [13], attachment disorders [14], and cognitive or psychiatric disorders during the child’s development [15].

Management of psychiatric disorders during the perinatal period remains a challenge. Indeed, psychotherapies can be hardly accessible [16], and there is a reluctance from both physicians and patients to use psychotropic treatments regarding their safety, in particular their potential impact on fetal development or their potential toxicity for the child during breastfeeding [17,18]. Therefore, 40 to 76% of pregnant women discontinue their psychotropic treatments before or during pregnancy [19], and it is estimated that antidepressants are five times more likely to be stopped in pregnant women [20]. In this context, it is crucial to develop the therapeutic armamentarium of psychiatric disorders during the perinatal period with treatments that are effective, safe, and well accepted by patients. Non-invasive brain neurostimulation (NIBS) techniques are emerging to treat peripartum psychiatric disorders, such as electroconvulsive therapy [21] or repetitive transcranial magnetic stimulation [22,23].

tDCS (transcranial direct current stimulation) is a NIBS technique that modulates brain activity using a weak intensity direct electric current delivered over short periods of time through two electrodes placed on the scalp: the anode, which facilitates spontaneous neuronal activity, and the cathode, which reduces spontaneous neuronal activity [24]. To date, there are several types of transcranial electrical stimulation (tES) [25]: tDCS uses a direct continuous electric current, as described above; transcranial alternating current stimulation (tACS) uses a sinusoidal alternating electric current [26]; and transcranial random noise stimulation (tRNS) uses a weak alternating current [27]. tES is increasingly used to treat a multitude of psychiatric disorders ranging from depression to schizophrenia and post-traumatic stress disorders (PTSD) [28,29,30]. In adults (pregnant women excluded) suffering from unipolar or bipolar depression, a recent meta-analysis based upon 23 randomized controlled trials (RCT) (1092 patients) demonstrated that tDCS (anodal stimulation of the left dorsolateral prefrontal cortex (F3), current intensity mostly at 2 mA, on 25 to 35 cm^2^ sponges, 20 to 30 min of stimulation over 5 to 20 sessions) had superior efficacy to placebo stimulation with a respective response rate of 33.3% versus 16.56% (OR: 2.28, 1.52 to 3.42), and a remission rate of 19.12% versus 9.78% (OR: 2.12, 1.42 to 3.16) [31]. Concerning patients suffering from schizophrenia, another recent meta-analysis based on 16 RCT (326 patients) demonstrated that adjuvant tDCS (anodal stimulation of the left dorsolateral prefrontal cortex (F3); cathodal stimulation at the left temporoparietal–parietal junction (T3P3); the current intensity at 2 mA; on sponges of 25 to 35 cm^2^; 20 min of stimulation over 5 to 40 sessions) was more effective than placebo stimulation on positive symptoms (standard mean difference (SMD): 0.17, 95% CI: 0.001 to 0.33), negative symptoms (SMD: 0.43, 95% CI: 0.11 to 0.75), and auditory hallucinations (SMD: 0.36, 95% CI: 0.02 to 0.70) [32]. For PTSD, tDCS was used—adjuvant to medication [33] or in association with virtual reality exposure [34]—to enhance fear extinction by stimulating the prefrontal cortex. tDCS is well tolerated, with minor and transient side effects; the most frequent of which are tingling of the scalp, itching, and fatigue [35,36]. The satisfactory tDCS safety, tolerance, and acceptability was also confirmed in children and adolescents [37].

It must be noted that among all tES techniques, tDCS is the most widely used and has the advantage of being easy to use (without anesthesia), safe (no serious side effects), and associated with a low dropout rate [35,38,39] inferior to 10% according to a meta-analysis based on 200 studies of 1000 patients with a total of 33,200 sessions [36]. tES could therefore constitute an interesting treatment for pregnant or breastfeeding women. However, data on the subject are still scarce. This systematic review aims to examine studies that explore the efficacy and safety of tES in the perinatal period, followed by a description of three case reports using tDCS during the perinatal period in cases of unipolar and bipolar depression and PTSD.

## 2. Methods

This systematic review of the literature was performed according to PRISMA systematic review guidelines [40]. Databases included MEDLINE (Pubmed) and ScienceDirect.

### 2.1. Eligibility Criteria

Studies were eligible if they strictly included humans. We selected studies that assessed tES–tDCS/tACS/tRNS—for a psychiatric disorder in women during the perinatal period. The studies also had to be written in English until 31 December 2021. In addition, we restricted our selection to peer-reviewed articles (excluding poster presentations, oral communications, letters to the editor, and book chapters). Search terms were selected to target our population (women during the perinatal period) and our subject of interest (treatment by tES) by using several keywords: ((«tDCS» OR «transcranial direct current stimulation» OR «tACS» OR “transcranial alternating current stimulation” OR “transcranial random noise stimulation” OR «tRNS») AND («pregnancy» OR «perinatal» OR «postnatal» OR «postpartum»)). The systematic review was not registered on PROSPERO.

### 2.2. Selection Methods

One reviewer (AL) searched using the terms cited above (in the eligibility criteria) to identify relevant studies that involved tES in women during peripartum. Titles and abstracts were screened by AL in order to remove duplicates. They were then assessed independently by two reviewers (AL and NN). Articles were excluded if both reviewers decided that an article clearly did not meet the criteria or if articles involved the use of tES for any reason other than a psychiatric disorder. In case of disagreement, the two reviewers had to reach a consensus. The same process was applied to the full-text versions of the remaining articles. 

### 2.3. Data Collection Process

For each case, we collected sociodemographic data, the psychiatric diagnosis motivating tES treatment, the stage of pregnancy at which tES was performed, the type of stimulation used, the stimulation procedure, as well as efficacy, safety, and tolerance data concerning the mother and child, including gynecological and obstetrical data. The risk of bias was not assessed, given the small number of articles involved.

### 2.4. Case Reports

After the systematic literature review, we described three clinical cases, including data on tDCS efficacy (with clinical data and psychometric scale scores (Montgomery–Åsberg Depression Rating Scale, [MADRS; [41]], Beck Depression Inventory-13 item [BDI-13;[42]], PTSD Checklist-5, [PCL-5; [43]], Clinical Administered PTSD Score [CAPS-5; [44]], EroQol-5D [ED-5Q; [45]], and the Montreal Cognitive Assessment score [MoCA; [46]]), tolerance (with adverse effects reported by patients) and safety for the mother and child (with data on pregnancy and delivery course, Apgar scores, term of birth, and birth weight of the child if applicable). We described the use of tDCS during the first-trimester pregnancy in a patient with bipolar depression (included in a research protocol [47]) and in a patient with post-traumatic stress disorder (included in an RCT, Clinical Trial n°NCT02900053). The third clinical case concerns the use of tDCS in postpartum depression in a nursing woman with tDCS parameters commonly used in the literature and described in the introduction (see [31]).

### 2.5. Statistical Analysis

Given the small number of studies available on the subject and their heterogeneity, only a descriptive analysis was performed to describe the results.

### 2.6. Ethics

All patients described in the case reports have given written consent for the use of their data.

## 3. Results

The initial search on PubMed and ScienceDirect was conducted until 31 December 2021 and provided 52 potentially eligible studies. No duplicates were found, and no article was excluded after assessing its title and abstract. After a full-text review of the 52 remaining reports, 45 were excluded: 20 studies had a different population, 17 reports did not assess tES, 4 studies were not written in English, and 4 did not present outcomes (clinical data on the efficacy or safety of tES in ante/postpartum women). As a result of the selection criteria, seven papers were selected (see Figure 1). In addition, three local case reports (described below) were added. Finally, seven papers were selected for the systematic review. 

### 3.1. Characteristics of the Studies

The results of our review are summarized in Table 1. All the studies concerned the use of tES during the gestation period in two indications: unipolar depression and schizophrenia.

Regarding depression, we found two cases reporting stimulation of the dorsolateral prefrontal cortex in women during the first trimester of pregnancy. In monotherapy, tDCS [49] and gamma-tACS [52] showed improvement in depression symptoms, with minor adverse effects during stimulation sessions–and with an improvement in cognitive scores with gamma-tACS. No negative consequences on pregnancy were reported. Our review of the literature also uncovered two open label trials. The study by Palm et al. [51] performed tDCS monotherapy in three pregnant women, ranging from the second to the third trimester. The treatment was effective with a reduction in depression scores. tDCS was well tolerated with an improvement in trail making test scores (executive functions). The second open-label trial included six pregnant women, ranging from the first to the second trimester of pregnancy [54]. tDCS treatment was adjunctive to psychotherapy, resulting in an improvement in depression symptoms at the two-week follow-up (phase 1). Adding additional tDCS sessions (phase 2) did not result in a significant reduction in depression scores. tDCS was well tolerated except for minor adverse effects, and cognitive scale scores were stable after phase 1 and improved after phase 2. No abnormalities were detected in the prenatal or neonatal period in any patient. Finally, we found a randomized controlled trial (RCT) [53] using tDCS monotherapy in 20 pregnant women (1:1), between the second and third trimester of pregnancy. Both groups improved in depression scores without superiority of tDCS over placebo. However, the remission rate was much higher in the tDCS group. tDCS was well tolerated, with only minor side effects reported during the sessions. The satisfaction rate was good, estimated at 87.5%, with a retention rate of 88%. Maternal and fetal monitoring showed no abnormalities. Only one woman delivered prematurely, and one child had an Apgar score < 8 at 1 min and normalized at 5 min. There were no between groups difference on any of the infant developmental-behavioural outcome indicators.

Regarding schizophrenia, there were two cases reporting the use of tDCS to reduce auditory verbal hallucinations (AVH). Both studies were inspired by the original study of Brunelin et al. [55], seeking to reduce the left temporo–parietal junction hyperactivity (T3-P3 cathodal stimulation) associated with AVH. Whether used alone [50] or as an add-on [48] therapy, tDCS was effective in reducing AVH. tDCS was well tolerated, with no adverse effects, and with no negative consequences on pregnancy.

### 3.2. Case Reports

Case report #1—**tDCS in a pregnant woman with type 2 bipolar depression**: Here, we present a 28-year-old female monitored for type 2 bipolar disorder. She had presented a first depressive episode at the age of 18 and had been hospitalized after a suicide attempt. She then suffered several depressive relapses. Several antidepressants were administered with no notable efficacy. A type 2 bipolar disorder diagnosis was finally made, as there were hypomanic periods between recurrent depressive episodes. Lithium treatment did not show any efficacy and was discontinued after a few months. The actual episode consisted of a moderate depressive relapse. Initially, treatment with lamotrigine 100 mg per day was prescribed, which was partially effective considering that the patient refused to increase the dosage. In view of the worsening depressive symptoms after a few months, the patient was offered adjunctive treatment by tDCS (see the following research protocol [47]). Before tDCS, the Montgomery–Åsberg Depression Rating Scale (MADRS) score was 32/60 [41], the Beck Depression Inventory-13 item (BDI-13) score was 21/39 [42], the EroQol-5D (ED-5Q) score was 40/100 [45], and the Montreal Cognitive Assessment (MoCA) score was 25/30 [46]. The tDCS protocol (Soterix^®^ device) performed included 15 sessions of 30 min over three weeks, that is one session per day with a current intensity of 2 mA (15 s fade-in and fade-out) delivered on 25 cm^2^ sponges, with anode placement on F3 and cathode on F4 (right and left dorsolateral prefrontal cortex, respectively). The patient reported some minor general side effects, such as fatigue and paresthesia. No mood swings were noted. Four days after the end of the tDCS treatment, i.e., one month after its beginning, the MADRS score was 15/60, BDI-13 score was 12/39, EQ-5D score was 50/100, and MoCA score was 26/30. The patient reported being pregnant a few days after the 1-month assessment. The tDCS treatment, therefore, took place between the first and third week of pregnancy. At two months and six months follow-up, MADRS and BDI-13 scores were, respectively, 18/60 and 12/39, and 13/60 and 11/39. The pregnancy went well with no complication reported. Labor was induced at 40 weeks and 5 days, a healthy baby was born (Apgar score 10/10 at one minute, birth weight 3.500 kg). One month later, the patient reported a stable mood, and there were no health issues regarding her child.

Case report #2—**Sham tDCS in a pregnant woman with PTSD**: We present a 34-year-old female referred by the victimology service for PTSD treatment by neuromodulation. The patient’s other notable medical history included severe fibromyalgia—treated with long-term analgesics—and hospitalizations in psychiatry wards for major depressive episodes with suicide attempts. At the time of her evaluation, the patient was receiving a treatment consisting of venlafaxine 75 mg per day, described as effective on her last depressive episode. The patient reported experiencing a traumatic event in 2019 when her nephew allegedly died suddenly at home of cardiac disease during a family event. Being a former caregiver, she allegedly performed cardiopulmonary resuscitation until the arrival of the fire department. A few days after this traumatic event, the first post-traumatic symptoms emerged before the full PTSD appeared two months later. We diagnosed chronic and non-dissociative PTSD. The Clinical Administered PTSD Score (CAPS-5) was 23/80 [44], the PTSD Checklist-5 (PCL-5) score was 50/80 [43], and the BDI-13 score was 12/39. The patient had no effective contraception as she just had her intra-uterine device removed due to intolerance. Considering that the patient did not express any pregnancy wish and considering that she was in search of a non-pharmaceutical treatment for her PTSD (no financial resources to pay for psychotherapy), we agreed on her request for tDCS (see Randomized Controlled Trial protocol on Clinical Trial n°NCT02900053). We proposed an adjunctive tDCS (or sham) treatment (Starstim^®^ device, NeuroElectrics, NIC software version 1.4) —while reading a traumatic script—according to the following parameters: a bi-encephalic set-up with the cathode placed on Fp2 and the anode placed on F3, the current intensity at 2 mA (30 s fade-in and fade-out with electrical stimulation at 2 mA between) or sham stimulation (only 30 s fade-in and fade-out) delivered on 20 cm^2^ sponges, 20 min of stimulation, two sessions per day (30 min between two sessions) over 5 consecutive days with an interval of 20 min between two stimulations, i.e., a total of 10 tDCS sessions. Before the first tDCS session, a negative pregnancy test had been performed. Sessions were well tolerated, with no serious adverse events. The patient reported some minor and transient adverse effects during sessions, such as tingling, difficulty concentrating, fatigue, scalp pain, itching, burning, and redness. One month after tDCS treatment, the patient reported a slight clinical improvement confirmed by psychometric scales with the following scores: CAPS-5 = 17/80, PCL-5 = 35/80 and BDI-13 = 8/39. At this one-month visit, the patient informed us that she was pregnant and that the first ultrasound dated the pregnancy to the weekend before the first tDCS session. Therefore, tDCS took place during the first week of pregnancy. As soon as the patient learned she was pregnant, she stopped all medication on her own so as not to take any risks regarding her pregnancy. The patient’s pregnancy progressed seamlessly with a standard gynecological and obstetrical follow-up. At the end of the pregnancy, and in the absence of analgesic treatment, the increasing and disabling pain of fibromyalgia motivated a delivery by cesarean scheduled at 39 weeks and 2 days of amenorrhea. The cesarean section went very well, without any complications, and the patient delivered a healthy baby boy with a birth weight of 3.120 kg for a height of 52 cm. Apgar scores were 10/10 at one and five minutes, and arterial cord pH was 7.31. One week after delivery, the child was healthy, and the mother was in remission. Indeed, post-traumatic symptoms gradually improved throughout the pregnancy. The patient was very satisfied with tDCS. At the end of the study, the unblinding revealed that the patient had received a sham (or placebo) stimulation.

Case report #3—**tDCS in a postpartum nursing woman**: Finally, we present a 28-year-old woman living with her husband and two children aged 19 months and 3 months. She had received brief psychological treatment in early childhood following her parents’ separation and had presented a first depressive episode at the age of 15 after a relationship break-up. Since then, the patient has described chronic depressive symptoms of variable intensity. She had sought medical attention before the birth of her second child, at four months of pregnancy, because of an intensification of depressive symptoms associated with suicidal thoughts. Treatment with sertraline 200 mg per day had then allowed a partial improvement in her condition. Given the persistence of depressive symptoms that had slightly worsened during the postpartum period, and given that breastfeeding prevented the multiplication of drug treatments, tDCS was finally offered at three months postpartum. tDCS protocol (Starstim^®^ device, NeuroElectrics, NIC software) included 15 sessions of 30 min over three weeks, i.e., one session per day with a current intensity of 2 mA (15 s fade-in and fade-out) delivered on sponges of 25 cm^2^ with the placement of the anode on F3 and the cathode on F4, followed by four maintenance tDCS sessions once a week. We observed a partial response to treatment after the tDCS course (MADRS scores range from 36/60 before tDCS treatment to 25/60 after tDCS treatment). The patient reported an improvement in mood and a decrease in attentional problems. The side effects reported were minor and transient and included mild fatigue, paresthesia of the scalp, and a transient low-intensity headache. She continued to breastfeed during treatment. One month later, when she was able to return to work, the patient reported a depressive relapse.

## 4. Discussion

The aims of our review were to identify available studies about tES during pregnancy and the postpartum period in order to investigate its efficacy and safety during the perinatal period. To our knowledge, several reviews have addressed this topic [56,57,58], but the present research has the advantage of including a larger number of studies independently of diagnosis and type of tES used. Thus, we highlighted the complete absence of available data on the use of tES in postpartum disorders and during breastfeeding, including postpartum depression. Evaluation of the effectiveness of neurostimulation techniques during the postpartum period seems to be particularly interesting. Indeed, some authors working on electroconvulsive therapy (ECT) seem to suggest a particular efficacy of neuromodulation during this period [59,60,61]. In this regard, other authors emphasized that the menstrual cycle, especially estrogens, has effects on cortical excitability [62], but to our knowledge, there is no study that addresses pregnancy’s effect on it.

Currently, the available data are sparse and limited to the use of tDCS or tACS in schizophrenia and unipolar depression during pregnancy. Our paper describes three different original case reports, including one case of tDCS treatment of bipolar depression, one of PTSD with sham (or placebo) tDCS, both during pregnancy, but also one case of tDCS usage in postpartum depression in a nursing woman. Presenting a sham (or placebo) tDCS case report may raise questions. In most studies, as in our case report, sham tDCS consists in emitting an electric field for a few seconds in order to imitate the physical sensations of active tDCS. There is now more and more evidence that repeated tES of a few seconds duration does have biological effects (for review and opinion, see [63]). It seemed legitimate to us to describe an original case report of sham tDCS stimulation in a pregnant woman to provide additional data on the safety and acceptability of tDCS in this specific population and to invite researchers to investigate the effects of placebo tDCS in their studies better. The results of our literature review, and of our three case reports, including obstetrical and fetal data, provide positive arguments for the use of tES in the perinatal period. Data suggest that tES allows, among other things, a reduction in depressive symptoms and AVH in schizophrenia. We did not find any paper on the use of tRNS in perinatal psychiatric disorders. This is not very surprising, considering this technique is very recent and still not widely used in clinical practice [26]. Beyond case reports or open-label trials, one of the gold-standard methods to rigorously evaluate the efficacy of a treatment is the randomized controlled trial [64], as in Vigod et al.’s study [53], which assessed the efficacy of tDCS versus placebo in 20 pregnant women. This study did not show the superiority of tDCS in reducing depression scales, but the rate of remission in the postpartum period was higher in the tDCS group than in the placebo group. These results are encouraging and should incite researchers to carry out other RCT on the subject, especially as the interpretation of Vigod et al.’s results is limited by the low power of the study, with a number of subjects that is probably too small to evaluate the effectiveness of tDCS [65].

Regarding safety, not all authors proposed the same monitoring of obstetrical and fetal parameters, but the available data report no consequences on pregnancy and birth, and all studies report good tolerance of tES, with minor and transient adverse effects well known for tDCS [36,39]. This good tolerance of tDCS is also found in Vigod et al.’s RCT [53], which proposed the most complete monitoring for mother and child and found a retention rate of 88%, similar to the rate found in the general population [35,38,39], making tES an effective treatment with few dropouts, including in pregnant women. Although little data are available at this time concerning the use of tES during pregnancy, it would be very unlikely that those treatments would have any adverse effects on a pregnant woman, embryo, or fetus. First, no serious general adverse effects have been demonstrated in several thousands of non-pregnant patients using tDCS or tACS [66]. Second, Shenoy’s team [48] emphasized that tDCS does not cause any significant changes in autonomic functions, ventilation rates, or core body temperature in healthy volunteers [67]. In addition, several authors point out that transcutaneous electrical nerve stimulation (TENS), with a strength of 100 mA, has been used safely in pregnancy for decades as pain relief during labor [49,68]. Compared to tES techniques, repetitive transcranial magnetic stimulation (rTMS) has been more widely studied in the perinatal period [57,58,69,70,71]. Kurzeck et al. [56] emphasized that follow-up examinations of children exposed to rTMS during pregnancy revealed no delay in cognitive or motor development [26]. They, therefore, suggested that electric field modeling could help predict current distribution during tDCS, as is already available for rTMS [24]. This electric field modeling has been estimated for ECT: Kibret et al. [72] discovered that the electric field used during ECT (which is far wider/larger than the electric field used in tES) reaching the fetal brain was most likely below the ICNIRP (International Commission for Non-Ionizing Radiation Protection) basic restrictions. These findings should really challenge clinicians and researchers about the use of tES as a treatment for psychiatric disorders during the perinatal period, especially considering that the current intensity is much lower than ECT or TENS. In the future, larger studies are therefore needed in order to assess the efficacy and safety of tES during the perinatal period, enrich available safety data, and specify the role/position of neurostimulation in the therapeutic algorithm of perinatal psychiatric disorders.

Regarding perspectives, studies about the experience of these women treated by tES are needed to evaluate the feasibility, acceptance, and perceptions of tES more precisely. Vigod et al. [53] reported patients’ views of treatment and identified global satisfaction and a good acceptance of neurostimulation for patients who refused medication. In order to complete these few data about women’s perceptions, qualitative studies could explore the experience of women treated by neurostimulation during the perinatal period and specify their motivations and eventual reluctance. In addition, efficacy data are limited to the effects of tES on psychiatric disorders and quality of life. Future studies could also include assessments of the effect of tES on pregnancy follow-up; on physical health, including long-term follow-up with assessments of neuro-cognitive development of children; or on the risk of postpartum depression when antepartum depression has been treated with tES. Finally, tES has the advantage of being potentially administered at home [73], considering that it is a portable machine with a lower cost than rTMS [74]. Home-based interventions seem to be particularly promising during the perinatal period, as women can encounter difficulties in following a treatment requiring daily trips to the hospital during their pregnancy or whilst having infants in their care. The realization of tDCS sessions at home, beyond the practical aspects, would give an active role to the patient in her care (self-determination or empowerment), joining the concept promoting of the subject’s freedom as a lever of the recovery process [75]. Thus, after rigorous monitoring of women and fetuses during tDCS sessions in their study and considering the safety of this treatment, [53] suggested that monitoring maternal blood pressure, heart rate, and fetal heart rate around the time of treatment in a clinic or at-home setting was reasonable. When considering the low cost of tDCS, this treatment could also be very interesting from a medical-economic point of view and medico-economic research on the subject would be welcome.

### Limitations

Although our review strived to be systematic, it obviously has some limits. First, the number of included studies is low. Second, these studies’ designs were heterogeneous and mainly consisted of case reports. We included one RCT, but the sample size was small and most likely involved in the lack of significant results. Third, we included studies that had very disparate pregnancy terms. Finally, we must consider the publication bias where the failures of tES, or the case of an adverse effect on the mother or child, would not have been published.

## 5. Conclusions

tES appears to be a safe and effective treatment for a number of psychiatric disorders during the perinatal period. Larger studies are needed to confirm these initial results, including during the postnatal period and using tACS or tRNS. All cases involving tES during the perinatal period—successful or unsuccessful, and with or without adverse effects on mother or child—should be reported to tES manufacturers: such feedback from the field can inform updated recommendations. Practitioners should also contribute to medical device safety surveillance efforts. In addition, research perspectives on the medico-economic benefits of tES, and its realization in patients’ homes, are to be investigated in the future.

## Figures and Tables

**Figure 1 jcm-11-04048-f001:**
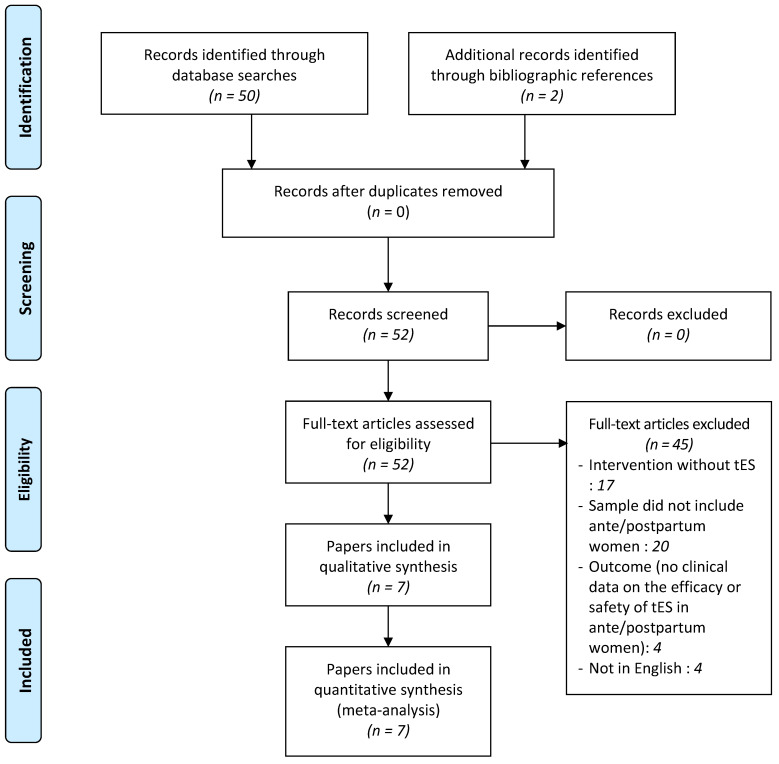
Adapted PRISMA flow diagram. *tES*: *transcranial electric stimulation*.

**Table 1 jcm-11-04048-t001:** Efficacy and tolerance of transcranial electric stimulation (tES) studies during perinatal period.

Studies	#Patients	Age (Years)	Term of Pregnancy (Gestational Weeks)	Disease Treated by tES	Type of tES (tES Device)	Anode Position	Cathode Position	Sponge Size	Stimulation Parameters	Results	Tolerance and Adverse Effects	Obstetrical and Fetal Data
[48] Case report	1	25	18	Schizophrenia	tDCS (pharmacotherapy adjuvant) TCT device	F3-FP1	T3-P3	Not stated	2 mA, 2 × 20 min/day, 3 h between two daily sessions, over 5 days (fade-in/fade-out not stated)	Progressive reduction in AHRS score from 29/42 to 22/42 (–24%) after treatment, then 2/42 (–93%) after one month follow-up.	No adverse effect occurred.	Repeat sonography showed healthy fetus (22 weeks) without any abnormalities, pregnancy was uneventful.
[49] Case report	1	23	6	Recurrent depressive disorder	tDCS (monotherapy) Neuroconn DC Stimulator Plus device	F3	F4	25 cm^2^	2 mA, 20 s fade-in/fade-out, 30 min/day, over 10 days	One month after the end of treatment, HAM-D reduced from 18 to 5 (–72%) and HAM-A reduced from 32 to 6 (–81%) and patient was in remission.	Minor adverse effect reported (3/10 sessions) during the fade-in phase: transient, mild burning sensations and fleeting experience of phosphenes	No information about fetal or obstetrical data.
[50] Case report	1	36	32	Schizophrenia	tDCS (monotherapy) Eldith DC-Stimulator (NeuroConn, Ilmenau, Germany)	F3	Tp3	Not stated	2 mA, 2 × 20 min/day, 3 h between two daily sessions, over 10 days (fade-in/fade-out not stated)	Changes in clinical scale scores at baseline, 2 weeks and 5 weeks follow-up were respectively: 18/49, 12/49 (–33%), 10/49 (–44%) for PANSS positive; 22/49, 23/49 (+5%), 24/49 (+9%) for PANSS negative; 39/112, 27/112 (–31%), 33/112 (–15%) for PANSS general (i.e., 15% reduction in total PANSS score); 27/42, 0/42 (–100%), 0/42 (–100%) for AHRS; 12/27, 8/27 (–33%), 7/27 (–41%) for CDSS; 45/100, 60/100 (+33), 71/100 (+57%) for GAF.	No adverse effect occurred.	Fetal examination via normal ultrasound at follow-up (35^th^ gestational week) revealed no changes or abnormalities. The spontaneous delivery of the healthy child occurred without any complications.
[51] Open label trial	3	23, 28 and 32	19 to 31	Major depressive disorder	tDCS (monotherapy) (device not stated)	F3	F4	Not stated	2 mA, 2 × 30 min/day, over 10 days (interval between two daily sessions not stated) ± 1 × 30 min/day over 10 days (fade-in/fade-out not stated)	Mean HAMD-21 total score reduced from 24.7 ± 10.7 to 15.7 ± 3.7 (–36%) after two weeks, then 7.0 ± 7.1 after four weeks (–72%). Mean baseline BDI-13 declined from 35.3 ± 12.5 to 12.0 ± 1.73 at week 2 (–66%), then to 11.0 ± 2.8 (–69%) at week 4. One patient achieved remission	tDCS was well tolerated, no adverse effect occurred. Mean baseline TMT-A was 25.0 ± 6.4 and changed to 23.3 ± 9.7 (–6.8%) in week 2, and to 18.5 ± 4.9 (–26%) in week 4. Mean baseline TMT-B was 81.0 ± 56.9 and sank to 69.3 ± 42.4 (–14%) in week 2, and to 40.5 ± 12.0 (–50%) in week 4.	No information about fetal or obstetrical data.
[52] Case report	1	38	6	Recurrent depressive disorder	Gamma-tACS (monotherapy) NeuroConn DC-Stimulator Plus	F3	F4	35 cm^2^	2 mA, 20 min, 40 Hz, 48,000 cycles, 9 sessions Offset at 1 mA without ramp-in/ramp-out	The scores at baseline, after 9 stimulations and then at 2 weeks follow-up were respectively 19 to 11 (–42%) then 10 (–47%) for HAMD-21; 24 to 12 (–50%) then 9 (–63%) for BDI; 26, 17 (–35%) then 15 (–42%) for PANAS negative affected scores; 15, 22 (+47%) then 30 (+100%) at PANAS positive affected scores After 3 months, the patient was in remission with a HAMD-21 score of 3 (–84%) and a BDI score of 7 (–71%).	Gamma-tACS was well tolerated with only mild phosphenes during stimulation and no further side effects. The scores at baseline, after 9 stimulations and then at 2 weeks follow-up were respectively 25s to 19s (–24%) then 15s (–40%) for TMT-A; and 82s to 50s (–40%) then 35s (–57%) for TMT-B.	No complications reported at 27 gestational weeks.
[53] RCT	20 (10:10)	26 to 43 (average age: 32.3 ± 4.15)	21 (median value)	Major depressive disorder	tDCS (monotherapy) Magstim device	F3	F4	35 cm^2^	2 mA (or sham), 1 × 30 min/day, 15 days over 3 weeks (fade-in/fade-out not stated)	At baseline, the total MADRS score was 23.5/60 (SD: 5.15) in the tDCS group, and 26.8/60 (SD: 7.48) in the sham-group. After, treatment, and using analysis of covariance, the estimated marginal mean MADRS score was 11.8/60 (SE: 2.66) in the tDCS group, and 15.4/60 (SE: 2.51) in the sham group (F = 0.97, p = 0.34). After treatment, the remission rate (MADRS score < 10) in the active and sham groups was 37.5% and 22% respectively, increased to 75% in the active group at 4 and 12 weeks postpartum versus 22% and 25% in the sham group.	The only side effects reported more than 3 times in either group was « buzzing » or « tingling » at the electrode site. There was no between-group difference in reported adverse effects. Two withdrawals in each group, for a retention rate of 88%, and the tDCS satisfaction rate was 87.5%.	Maternal heart rate, blood pressure and fetal monitoring were all within normal limits in both groups. No abnormalities noted on continuous fetal monitoring for women ≥ 24 weeks. No serious pregnancy complications reported in either group. Mean gestational age at birth was 39.0 week ± 1.4 in tDCS, and 38.9 week ± 1.1 in sham-control. Mean birth weight was 7.0 lbs ± 0.54 and 7.1 lbs ± 1.2 in tDCS and sham groups respectively. There was 1 child in each group with an Apgar score less than 8 at 1 min after birth and no infants with an Apgar score less than 8 at 5 min after birth. One infant in the tDCS group had a spontaneous preterm birth (36 weeks and 5 days gestation) with no known further sequelae. There were no other neonatal complications. There were no differences between groups on any of the infant developmental-behavioural outcome indicators.
[54] Open label pilot trial	6	23 to 43	12 to 33	Recurrent depressive disorder	tDCS (psychotherapy adjuvant) Eldith-DC-stimulator (NeuroCareGroup, Munich, Germany)	F3	F4	35 cm^2^	2 mA, 15 s fade-in/fade-out, 2 × 30 min/day, 10 days, accompanied by standard group psychotherapy sessions twice a week for 90 min each (phase 1) (interval between two daily sessions not stated) ± 1 × 30 min/day over 10 days (n = 4) (phase 2)	In phase 1 (n = 6), mean HAMD-21 total score decreased from 22.50 ± 7.56 to 13.67 ± 3.93 (–39%) after two weeks: two patients were responders defined by a 50% reduction of the HAMD-21 total score. Mean BDI-13 total score decreased from 26 ± 13.90 to 11.17 ± 5.46 (–57%) after two weeks: two patients were responders, and one patient was in remission defined by a HAMD-21 total score ≤ 7. CGI improved by 28.57%. Significant improvement of the WHOQOL “Psychological health “ sub-score. For patients who have completed phases 1 + 2 (n = 4), no significant reduction was found in HAMD-21 and BDI-13 sum scores after the phase 2.	The tDCS was well tolerated with no serious adverse effects. Patients reported the following transient adverse effects in association with tDCS: mild headache, phosphenes, and feeling of itching. The mean scores for CRQ questions 1, 2 and 3 were 19.8, 14.6 and 1.5, respectively. The TMT-A/B scores did not change during the phase 1. For patients who have completed phases 1 + 2 (n = 4), only TMT-A showed significant reduction (baseline: 25.79 ± 4.91; after phase 2: 19.33 ± 3.20).	Irregularities of fetal and maternal health were not detected during prenatal and neonatal periods in regularly performed check-ups in accordance with the obstetricians, including fetal heart rate measurement.
Current report #1	1	28	3 to 5	Bipolar type 2 depression	tDCS (pharmacotherapy adjuvant) Soterix device	F3	F4	25 cm^2^	2 mA, 15 s fade-in/fade-out, 1 × 30 min/day, 5 days per week, over 3 consecutives weeks	Reduction in MADRS scores from 32/60 to 15/60 (–53%) four days after the end of treatment, then 18/60 (–43%) and 13/60 (–59%) at 2 months and 6 months respectively. Improvement in BDI-13 scores from 21/39 to 12/39 (–43%) four days after the end of treatment, then 12/39 (–43%) and 11/39 (–48%) at 2 months and 6 months respectively. Improved quality of life (EroQol-5D scores from 40/100 to 50/100 (+25%)).	tDCS was well tolerated without severe adverse effect. The patient reported paresthesia of the scalp during the tDCS sessions and asthenia after the sessions. MoCA scores improved from 25/30 to 26/30 after treatment.	The pregnancy went well with an induced labor at 40 weeks and 5 days. The baby was born healthy (Apgar score 10/10 at one minute, birth weight: 3.500 kg)
Current report #2	1	34	3	PTSD	Sham (placebo) tDCS during the reading of a traumatic script (pharmacotherapy adjuvant) Starstim device (NeuroElectrics, Barcelona, Spain) NIC software	F3	Fp2	20 cm^2^	Sham stimulation, 30 s fade-in/fade-out, 2 sessions/day, over 5 consecutive days, 30 min between two daily sessions	One month after tDCS treatment, reduction in CAPS-5 scores from 23/80 to 17/80 (–26%), PCL-5 from 50/80 to 35/80 (–30%) and BDI-13 from 12/39 to 8/39 (–33%). PTSD in remission at delivery.	The tDCS was well tolerated with no major adverse effect, the patient reported minor and transient adverse effects during tDCS sessions such as tingling, difficulty concentrating, fatigue, scalp pain, itching, burning or redness.	The pregnancy went well with a scheduled cesarean delivery at 39 weeks and 2 days of amenorrhea for disabling fibromyalgia pain. Birth of a healthy child with a birth weight of 3.120 kg, a birth height of 52 cm, Apgar scores of 10/10 at one and five minutes, arterial pH of 7.31.
Current report #3	1	28	Breastfeeding woman during the postpartum period	Recurrent depressive disorder	tDCS (pharmacotherapy adjuvant) Starstim device (NeuroElectrics, Barcelona, Spain) NIC software	F3	F4	25 cm^2^	2 mA, 15 s fade-in/fade-out, 1 × 30 min/day, 5 days per week, over 3 consecutives weeks, then 4 weekly maintenance tDCS sessions	Reduction of MADRS score from 36/60 to 25/60 (–30%) after tDCS treatment with a relapse of depression at 1 month.	The side effects reported were minor and transient, included mild fatigue, paresthesias of the scalp, and a headache of low intensity.	Not applicable

Scores are expressed as absolute numbers or mean ± standard deviation. % indicates the evolution of the clinical scale scores in comparison with the baseline. AHRS: Auditory Hallucination Rating Scale; BDI: Beck Depression Inventory; CAPS-5: Clinical-Administered PTSD Scale for DSM-5; CDSS: Calgary Depression Scale for Schizophrenia; CGI: Clinical Global Impression; CRQ: Comfort Rating Questionnaire; EroQuol-5D: European Quality of Life 5 dimensions; GAF: Global Assessment Functioning; HAMD: Hamilton Rating Scale for Depression; HAM-A/B: Hamilton Depression Rating Scale; Hz: Hertz; mA: milliampere; min: minute; MoCA: Montreal Cognitive Assessment; PANAS: Positive And Negative Affect Schedule; PANSS: Positive And Negative Syndrome Scale; PCL-5: PTSD Checklist for DSM-5; PTSD: Post-Traumatic Stress Disorder; s: second; SD: Standard Deviation; SE: Standard Error; tACS: transcranial Alternating Current Stimulation; tDCS: transcranial Direct Current Stimulation; tES: transcranial Electric Stimulation; TMT-A: Trail Making Test parts A; TMT-B: Trail Making Test parts B; WHOQOL: WHO Quality of Life-BRE.

## Data Availability

Not applicable.

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
