# Peer review of "Efficacy and Safety of Transcranial Electric Stimulation during the Perinatal Period: A Systematic Literature Review and Three Case Reports"

_jcm, 2022, doi:10.3390/jcm11144048_

Round 1

Reviewer 1 Report

Title

-Modify the title indicating that this is a review study about pregnant mothers

Introduction

-I suggest including some studies assessing the efficacy of transcranial electric stimulation on psychiatric disorders in other populations.

Methods

-Please describe what kind of analyses was performed to get the results reported, Was the analysis quantitative or qualitative?

Results

-I suggest including a brief description of case studies in the methods section, psychometric tests, and how you selected the treatment or type of stimulation. It looks unformal in the current format.

Author Response

Reviewer #1

We thank very much the reviewer for the positive appreciation of our work.

  1. Title: Modify the title indicating that this is a review study about pregnant mothers.

We thank the reviewer for his suggestion. However, our study and clinical cases studying tES in the perinatal period included the pregnancy period but also the postpartum and breastfeeding period. We felt that using the "pregnant mothers" term in the title was too restrictive for our study population (even though the results of the review are focused on this population) and reduced the originality of our work. We hope that the reviewer will understand and accept our position on this point.

  1. Introduction: I suggest including some studies assessing the efficacy of transcranial electric stimulation on psychiatric disorders in other populations.

We thank the reviewer for his suggestion. To improve understanding of the practice of tES in the psychiatric population, we have added the following sentences (lines 105 to 125):

“In adults (pregnant women excluded) suffering from unipolar or bipolar depression, a recent meta-analysis based upon 23 randomized controlled trials (RCT) (1,092 patients) demonstrated that tDCS (anodal stimulation of the left dorsolateral prefrontal cortex (F3), current intensity mostly at 2mA, on 25 to 35 cm2 sponges, 20 to 30 minutes of stimulation over 5 to 20 sessions) had a superior efficacy to placebo stimulation with a respective response rate of 33. 3% versus 16.56% (OR: 2.28, 1.52 to 3.42), and a remission rate of 19.12% versus 9.78% (OR: 2.12, 1.42 to 3.16) (Razza et al., 2020). Concerning patients suffering from schizophrenia, another recent meta-analysis based on 16 RCT (326 patients) demonstrated that adjuvant tDCS (anodal stimulation of the left dorsolateral prefrontal cortex (F3), cathodal stimulation at the left temporoparietal-parietal junction (T3P3), current intensity at 2mA, on sponges of 25 to 35 cm2, 20 minutes of stimulation over 5 to 40 sessions) was more effective than placebo stimulation on positive symptoms (standard mean difference (SMD): 0.17, 95% CI: 0.001 to 0.33), negative symptoms (SMD: 0.43, 95% CI: 0.11 to 0.75) and auditory hallucinations (SMD: 0.36, 95% CI: 0.02 to 0.70) (Cheng et al., 2020). For PTSD, tDCS was used – adjuvant to medication (Ahmadizadeh et al., 2019) or in association with virtual reality exposure (Van’t Wout et al., 2019) - to enhance fear extinction by stimulating the prefrontal cortex. tDCS is well tolerated, with minor and transient side effects, the most frequent of which are tingling of the scalp, itching and fatigue (Aparício et al., 2016)(Bikson et al., 2016). The satisfactory tDCS safety, tolerancy and acceptability has also been confirmed in children and adolescents (Buchanan et al., 2021).”

  1. Methods: Please describe what kind of analyses was performed to get the results reported. Was the analysis quantitative or qualitative?

We thank the reviewer for his vigilance. We have added a brief description of the kind of analyses performed (lines 187 to 189).

“Statistical analysis

Given the small number of studies available on the subject and their heterogeneity, only a descriptive

analysis was performed to describe the results.”

  1. Results: I suggest including a brief description of case studies in the methods section, psychometric tests, and how you selected the treatment or type of stimulation. It looks unformal in the current format.

We agree with the reviewer and thank him for his vigilance. We have added the following sentences (lines 171-185):

“Case reports

After the systematic literature review, we described three clinical cases including data on tDCS efficacy (with clinical data and psychometric scale scores (Montgomery-Åsberg Depression Rating Scale, MADRS (Montgomery & Asberg, 1979), Beck Depression Inventory-13 item, BDI-13 (Beck & Beamesderfer, 1974), PTSD Checklist-5, PCL-5 (Ashbaugh et al., 2016), Clinical Administered PTSD Score, CAPS-5 (Weathers et al., 2013),EroQol-5D, ED-5Q (Chevalier & de Pouvourville, 2013) and the Montreal Cognitive Assessment score, MoCA (Nasreddine et al., 2005)), tolerance (with adverse effects reported by patients) and safety for the mother and child (with data on pregnancy and delivery course, Apgar scores, term of birth, birth weight of child if applicable). We described the use of tDCS during the first trimester pregnancy in a patient with bipolar depression (included in a research protocol (Sauvaget et al., 2020)) and in a patient with posttraumatic stress disorder (included in a RCT, Clinical Trial n°NCT02900053). The third clinical case concerns the use of tDCS in a postpartum depression in a nursing woman with tDCS parameters commonly used in the literature and described in the introduction (see Razza et al., 2020).”

Reviewer 2 Report

Interesting read especially in the context that many pregnant women ask me in clinical practice (at least to the reviewer´s experience )whether the use of tDCS (and TMS) is safe in perinatal period.
The review is presented well, strived to be systematic, according to PRISMA guidelines and the weaknesses are reasonably presented. The authors selected 7 articles and 3 localised case reports based on their search method and come to the conclusion that the application of tDCS during the perinatal period, based on the current available reporting/data, appears to be safe and effective and discuss the potential benefit of the potential applications of this method in a home environment by the patients themselves. 
tDCS is an increasingly popular method on the field of neuromodulation with a rising potential - many pharmacological approaches cannot be utilised during perinatal period due to their potential adverse effects, and tDCS appears to represent a safe and easy to use alternative/augmentation - articles such as this one are important in this consideration. 

I have several suggestions for the author in order to improve their article: 

1. English is not my first language but I have spotted some typos throughout the text, consider a professional spelling/grammar check for the article。

2. Consider adding study limitations part to have its separate title
3. Although tDCS appears to be safe in the perinatal period, as the authors indeed mention, the overal population sample from the selected studies is small. The authors should stress more in the abstract and conclusion for the need to report ANY adverse effect that occurs in this type of stimulation during the perinatal period. 

Author Response

Reviewer #2

We thank very much the reviewer for the positive appreciation of our work.

  1. English is not my first language, but I have spotted some typos throughout the text, consider a professional spelling/grammar check for the article.

We apologize for the inconvenience. The article was proofread by a fluent English speaker.

  1. Consider adding study limitations part to have its separate title.

We thank the reviewer for his rigorous review. As requested, we have individualized a "limitations" sub-section in the discussion. We hope this meets the reviewer's expectations.

  1. Although tDCS appears to be safe in the perinatal period, as the authors indeed mention, the overal population sample from the selected studies is small. The authors should stress more in the abstract and conclusion for the need to report ANY adverse effect that occurs in this type of stimulation during the perinatal period. 

We thank the reviewer for this important suggestion. As required, we have added the following sentences:

  • All cases involving tES during the perinatal period — successful or unsuccessful, and with or without adverse effects on mother or child — should be reported to tES manufacturers: such feedback from the field can inform updated recommendations. Practitioners should also contribute to medical device safety surveillance efforts. (Lines 461-465)
  • any adverse effects on the mother or child should be reported (line 51, abstract)

Round 2

Reviewer 1 Report

I am satisfied with the changes made to the manuscript.